# The Longitudinal Pediatric Data Resource: Facilitating Longitudinal Collection of Health Information to Inform Clinical Care and Guide Newborn Screening Efforts

**DOI:** 10.3390/ijns7030037

**Published:** 2021-06-30

**Authors:** Amy Brower, Kee Chan, Michael Hartnett, Jennifer Taylor

**Affiliations:** American College of Medical Genetics and Genomics (ACMG), 7101 Wisconsin Avenue Suite 1101, Bethesda, MD 20814, USA; kchan@acmg.net (K.C.); mhartnett@acmg.net (M.H.); jtaylor@acmg.net (J.T.)

**Keywords:** newborn screening, research, long-term follow-up, NBSTRN, LPDR, RUSP

## Abstract

The goal of newborn screening is to improve health outcomes by identifying and treating affected newborns. This manuscript provides an overview of a data tool to facilitate the longitudinal collection of health information on newborns diagnosed with a condition through NBS. The Newborn Screening Translational Research Network (NBSTRN) developed the Longitudinal Pediatric Data Resource (LPDR) to capture, store, analyze, visualize, and share genomic and phenotypic data over the lifespan of NBS identified newborns to facilitate understanding of genetic disease and to assess the impact of early identification and treatment. NBSTRN developed a consensus-based process using clinical care experts to create, maintain, and evolve question and answer sets organized into common data elements (CDEs). The LPDR contains 24,172 core and disease specific CDEs for 118 rare genetic diseases, and the CDEs are being made available through the NIH CDE Repository. The number of CDEs for each condition average of 2200 with a range from 69 to 7944. The LPDR is used by state NBS programs, clinical researchers, and community-based organizations. Case level, de-identified data sets are available for secondary research and data mining. The development of the LPDR for longitudinal data gathering, sharing, and analysis supports research and facilitates the translation of discoveries into clinical practice.

## 1. Introduction

Each of the approximately 4 million babies born in the United States each year receives neonatal screening for over thirty-six rare genetic diseases [1]. Newborns who screen positive undergo a series of confirmatory tests, and a subset ultimately gets a diagnosis. Screening and short-term follow-up occur within the state-based public health system while diagnosing, treating, and managing sick newborns occurs in pediatric care settings. This series of hand-offs from prenatal care to public health to clinical care creates a unique opportunity to capture crucial longitudinal health information for infants diagnosed through NBS. However, there is no national system to collect, analyze and share this information [2].

Newborn screening is a system of interconnected activities that begin before a baby is born. This multi-stakeholder system’s key components include prenatal education, neonatal screening, short-term follow-up and diagnosis, and clinical care and management. Each year at least 12,905 newborns are diagnosed with a condition through NBS, and the majority enter life-long care management [3]. However, the success of NBS has been measured by the number of newborns identified with a screened condition and not by health outcomes in diagnosed newborns. Our inability to quantify the net benefit of NBS by improved health outcomes may be a consequence of no national approach to longitudinal data collection, analysis, and dissemination. In the past ten years, a federal advisory committee, the Advisory Committee on Heritable Disorders in Newborns and Children (ACHDNC), has recognized the importance of long-term follow-up (LTFU). ACHDNC authored several publications defining the key features and components of NBS longitudinal data collection efforts, describing questions to be addressed, and identifying potential sources of data [4,5,6]. They also defined the goal of LTFU as assuring the best possible outcome for individuals with disorders identified through NBS and identified three key features and four central components of LTFU (Figure 1).

The phrase “newborn screening long-term follow-up” applies both to the longitudinal treatment and management of diagnosed newborns as well as the longitudinal data collection of health information to inform assessments of outcomes. The Newborn Screening Saves Lives Reauthorization Act addressed LTFU as “follow-up activities, including those necessary to achieve rapid diagnosis in the short-term, and those that ascertain long-term management outcomes and appropriate access to related services.” [7]. Both activities are essential to advance NBS. The long-term clinical management of newborns diagnosed with a condition through newborn screening is necessary to ensure that we achieve the best possible outcomes for these infants. The ability to capture clinical information early in the clinical course of a disease, even before clinical symptoms appear, advances disease understanding. Longitudinal data collection also helps to establish the efficacy of new treatments and management approaches, informs the community at large about the value of early identification and treatment through newborn screening, and identifies areas for improvement in disease management throughout the lifespan. Also, longitudinal data collected on children diagnosed through population-based screening offers significant potential to advance scientific and clinical understanding of NBS conditions and document the life-saving benefit of NBS. It enables individual and group level analysis from a data set with unbiased ascertainment. Longitudinal Data also has the capability to capture information on family experience, improving the capacity at which clinical experts can care for their patients and their families. In this paper, we describe efforts to facilitate the longitudinal capture and dissemination of health information on newborns diagnosed with a condition through NBS.

## 2. Materials and Methods

### 2.1. Newborn Screening Translational Research Network

The Eunice Kennedy Shriver National Institute of Child Health and Human Development (NICHD) Hunter Kelly Newborn Screening Research Program was created to support investigations and newborn screening innovations [8]. Recent efforts have explored the use of genomics in the neonatal period, conducted prospective pilots of conditions that are candidates for nationwide screening to evaluate the clinical benefit, and developed novel screening technologies for candidate conditions. The American College of Medical Genetics and Genomics (ACMG) plays a vital role in these ground-breaking efforts by leading the NICHD funded Newborn Screening Translational Research Network. NBSTRN is a key component of the Hunter Kelly Newborn Screening Research Program and began as an effort to engage a variety of stakeholders across the NBS system [9]. NBSTRN has now matured into a dynamic and committed network comprised of researchers, healthcare professionals, state NBS programs, families, and advocacy groups. The NBSTRN team develops tools to facilitate the discovery and validation of novel technologies to screen and diagnose disease, pilot new technologies and treatments, describe the ethical, legal, and social implications of NBS research, and collect longitudinal health and genomic data. 

NBS in the US is a multi-component, multi-stakeholder system of prenatal education, hospital and state-based public health laboratory screening, clinician and state-based laboratory confirmation and diagnosis, clinical treatment and management, and health outcome analysis. The NBSTRN data tools, resources, and expertise are designed to facilitate all stakeholders’ efforts and leverages each component of NBS to advance research. As a translational research network, NBSTRN was conceptualized to develop and share resources and infrastructure to support NBS researchers. The collective research infrastructure provides mechanisms for an unbiased understanding of rare genetic conditions across the lifespan, population-based pilot testing of new NBS tests, new technology development and application, and new therapeutics development. This research infrastructure also strengthens and enables the expansion of NBS programs in a more systematic fashion. 

### 2.2. Development of Common Data Elements (CDEs)

ACMG also operates the National Coordinating Center (NCC) for the Regional Genetics Networks (RGNs) [10]. In 2015, two representatives from each of the seven RGNs formed a joint committee with an expert NBSTRN workgroup called the Clinical Integration Group (CIG). The joint committee of experts developed sets of questions and answer choices, called common data elements (CDEs), for NBS conditions that are part of the Recommended Uniform Screening Panel (RUSP) [11]. In newborn screening, the use, and development of CDEs are focused on facilitating data collection, standardization, sharing, aggregation, analysis, and dissemination. Combining data sets is especially important in newborn screening because most conditions are rare and accumulating enough subjects to have statistical power is often a barrier to understanding health outcomes and the benefits of early identification and treatment. 

The goal was to create a catalog of CDEs to enable investigators and public health teams to systematically collect, analyze, and share data across the NBS community. The development of the CDEs was guided by ACHDNC’s publications and Follow-Up and Treatment Subcommittee discussions [12]. The CDEs were designed to be applicable for both research and public health and contained core and disease specific CDEs. The NBS CDE sets’ potential uses include natural history studies, hypothesis-driven efforts, surveillance, outcomes, quality assurance and improvement, and national benchmarks. A clinician may use the NBS CDE sets to describe the clinical course of the NBS identified condition in asymptomatic patients to facilitate a new understanding of the disease. At the same time, another may explore the relationship between genomic variants characterized at birth and later outcomes like lung function in cystic fibrosis. A public health partner may use the NBS CDE sets to describe the relationship between service delivery and treatment methods to define optimal follow-up care plans for children with an NBS condition.

NBSTRN was charged with creating a data tool using the resulting consensus based, standardized CDEs, organizing the CDEs into case report forms, and making the forms available through a secure online portal facilitating data collection and management. NBSTRN also worked with subject matter experts to develop CDE sets for conditions that are candidates for newborn screening. CDE dissemination was accomplished through a collaboration with the National Institutes of Health (NIH) National Library of Medicine (NLM) and NBSTRN via the NIH CDE Repository (Repository) [13]. Currently, this repository catalogs 26,518 elements across 16 classifications, with multiple NIH Institutes and efforts being represented. The Repository allows researchers to build data collection instruments from shared CDEs and contribute generated data elements. To foster the use of standardized CDEs, NBSTRN deposits the question-and-answer sets in the NICHD module of the Repository. This repository has been designed to provide access to structured human and machine-readable definitions of data elements that have been recommended or required by NIH Institutes and Centers and other organizations for use in research and other purposes.

### 2.3. Longitudinal Pediatric Data Resource

The neonatal screening of newborns each year leads to the diagnosis of infants with a genetic condition that requires referral to clinical care and, in most cases, lifelong management. This unselected cohort of newborns reflects our nation’s racial, geographic, economic, and educational diversity. This may be the perfect cohort to help advance disease understanding because although every newborn receives essentially the same screen, other factors vary, including treatment choice and disease course. Also, many of the screened conditions have comorbidities, including intellectual disabilities. These children receive various interventions that could be tracked and analyzed to identify critical periods of development and intervention. Because the NBS system in the United States effectively screens over 99% of newborns, it can provide a unique platform for understanding rare diseases and lifelong outcomes. The process of neonatal screening followed by a coordinated transition to clinical care facilitates collecting health information beginning just hours after birth. And because the majority of NBS conditions require life-long care and management, we have the opportunity to conduct prospective, longitudinal natural history studies on a population basis with unbiased ascertainment. 

Over the last decade, NBSTRN has developed a data tool, the Longitudinal Pediatric Data Resource (LPDR), to support several landmark natural history studies that have contributed to a better understanding of the etiology, pathophysiology, and phenotypic heterogeneity of NBS conditions and provided an assessment of health outcomes [14]. The LPDR is a suite of tools to collect, analyze, visualize, store, and share genomic and phenotypic data. The LPDR is housed within a Federal Information Security Management Act (FISMA) of 2002 (Public Law 107–347) Moderate cloud environment, is available for use by all NBSTRN stakeholders, and is designed to share both de-identified, case-level, and aggregate data sets and foster secondary use of accumulated data [15]. Data stored in the LPDR is available to the research and NBS community, and the use of de-identified, case-level data requires registration as an NBSTRN user. Depositing case-level data into the LPDR and requesting the use of complete data sets that include personal health information (PHI) involves the execution of a Data Sharing Agreement (DSA). The LPDR also facilitates data sharing and data standardization with 3rd party databases, including the NIH National Center for Biotechnology Information (NCBI) database of Genotypes and Phenotype (dbGaP) and the NLM NIH Code Repository (https://cde.nlm.nih.gov, accessed on 12 April 2021). The LPDR provides data dictionaries to create electronic data entry forms and features de-identified, case-level data sets for data mining. The secondary use of accrued LPDR data may help to (1) establish the efficacy of new treatments and management approaches, (2) inform the community about the value of early identification and treatment through newborn screening, and (3) identify areas for improvement in disease management throughout the lifespan.

The LPDR is accessible through the NBSTRN website at nbstrn.org and operates in a cloud environment that utilizes the NIH Science and Technology Research Infrastructure for Discovery, Experimentation, and Sustainability (STRIDES) Initiative and Amazon Web Services (AWS) [16]. The NBSTRN team follows federal policies for information security protections outlined under FISMA. These policies outline requirements for developing and operating information technology (IT) systems based on data type, data use, and risk impact. There are three security categories—low, moderate, or high. ACMG received authority to operate (ATO) the NBSTRN website, tools, and resources at the moderate level, and this enables the collection, storage, and use of personal health information (PHI). ACMG submits a quarterly review of the system’s security to NICHD, and this provides the NBSTRN network with an approved infrastructure to facilitate the protected and secure collection, analysis, and dissemination of NBS research data. 

## 3. Results

### 3.1. CDE Sets

A total of 24,172 core and disease specific CDEs have been developed for use in longitudinal health information data collection efforts for 118 conditions that are part of, or candidates for, nationwide NBS. The joint committee created 15,685 CDEs for 46 RUSP conditions. The NBSTRN developed 8487 additional disease specific CDEs for 72 diseases for use by researchers conducting natural history studies or pilots of conditions recently added to, or candidates for, the RUSP (Appendix A). The CDEs were used to create data dictionaries and electronic case report forms (eCRFs) organized chronologically to match a subject’s longitudinal care record over time (Figure 2). Most research projects use REDCap for data collection and formats the CDEs into dropdown menus using REDCap (Figure 2) [17]. The CDEs were made available to the NLM for dissemination via the NIH CDE Repository, NICHD Module, and integration into EHR standards for the US and the NBS Coding and Terminology Guide [18]. A data almanac was created that contained semantic definitions, annotations, and standardization to improve the data collected by each investigator. 3745 of the 15,685 (24%) of the CDEs currently have definitions and, where applicable, annotated with Logical Observation Identifiers Names and Codes (LOINC), Systematized Nomenclature of Medicine–Clinical Terms (SNOMED CT), and the International Classification of Diseases, Ninth or Tenth Revision (ICD-9 or 10) codes [19]. Disease-specific and study-specific CDE sets are available online at https://nbstrn.org/tools/lpdr (accessed on 12 April 2021). CDE sets are organized into REDCap formatted data dictionaries and text files. The LPDR enables users to adopt the consensus based CDEs, generate case report forms, and suggest additional CDEs for new conditions, technologies, and treatments.

### 3.2. LPDR: Phenotypic and Genomic Data

Since its launch in 2013, the LPDR has been utilized by several research teams, either conducting longitudinal studies of both RUSP and candidate conditions, exploring the use of genome sequencing in the newborn period, and executing NBS pilots of candidate conditions. The LPDR has 312 registered users, 151 individuals approved to enter or upload data, 23 established longitudinal projects, ~12 Million data points, and 8842 subjects with an average of 3.62 data collection time points per subject. The LPDR has data displays that describe the types of data and populations available for secondary use (Figure 3). Our data governance and data sharing policies provide qualified researchers access to these data sets. Case-level, de-identified data sets from 8 projects are featured within the LPDR and are available for secondary use by the research community. Data dashboards for the eight featured projects highlight critical aspects of the accumulated data and population. Thirteen active and completed projects provide a representative sample of the type of efforts that utilize the LPDR (Table 1).

## 4. Discussion

Longitudinal care management of newborn screen identified individuals coupled with longitudinal capture of health information are vital cornerstones of a successful NBS system in the United States. The NBSTRN led an effort to develop the LPDR, a suite of tools and resources to facilitate longitudinal data capture using question and answer sets developed by clinical experts. This effort resulted in the identification of key data elements for long-term follow-up (LTFU) efforts conducted by state NBS programs. These data elements could also be used by researchers, advocates, policy makers, and clinicians to evaluate the health outcomes of newborns identified by NBS. The LPDR has been utilized in a variety of efforts, including a ten-year effort to collect, analyze and disseminate health information on individuals with one of 42 RUSP conditions in 30 clinical sites located in 22 states, multi-state pilots of 4 conditions collectively screening over 1.2 M births, use of genome sequencing in four cohorts of newborns including neonatal intensive care unit and healthy, studies expanding the diagnostic window of NBS both beyond and before the neonatal period, and efforts to consolidate disparate patient registries into a single data dictionary that supports pilots. As the data sets contained in the LPDR grow, the NBS community can utilize this information to inform the care and management of newborns identified with a condition through NBS, guide policy and funding decisions, assess the benefit of NBS, and further our understanding of these rare genetic conditions.

To facilitate the utilization and application of LPDR, NBSTRN has displayed a set of LPDR studies and data visualization on the website www.nbstrn.org (accessed on 12 April 2021) (see Figure 4). By becoming a member of NBSTRN, patrons can have access to the de-identified case-level data, conduct secondary data analysis and explore additional research questions such as how often individuals with metabolic diseases have hearing loss. 

## Figures and Tables

**Figure 1 IJNS-07-00037-f001:**
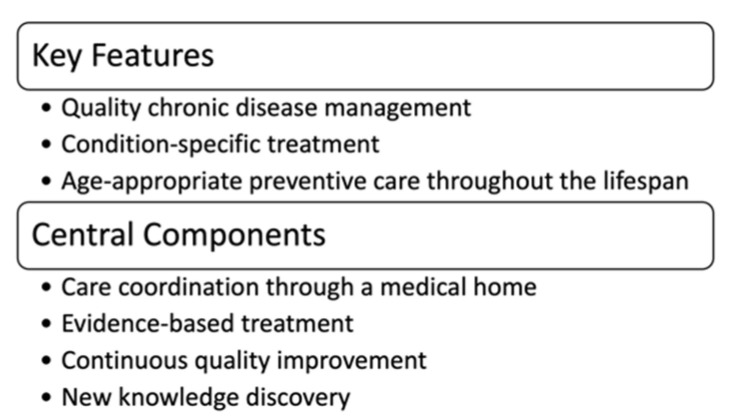
ACHDNC Statement on LTFU.

**Figure 2 IJNS-07-00037-f002:**
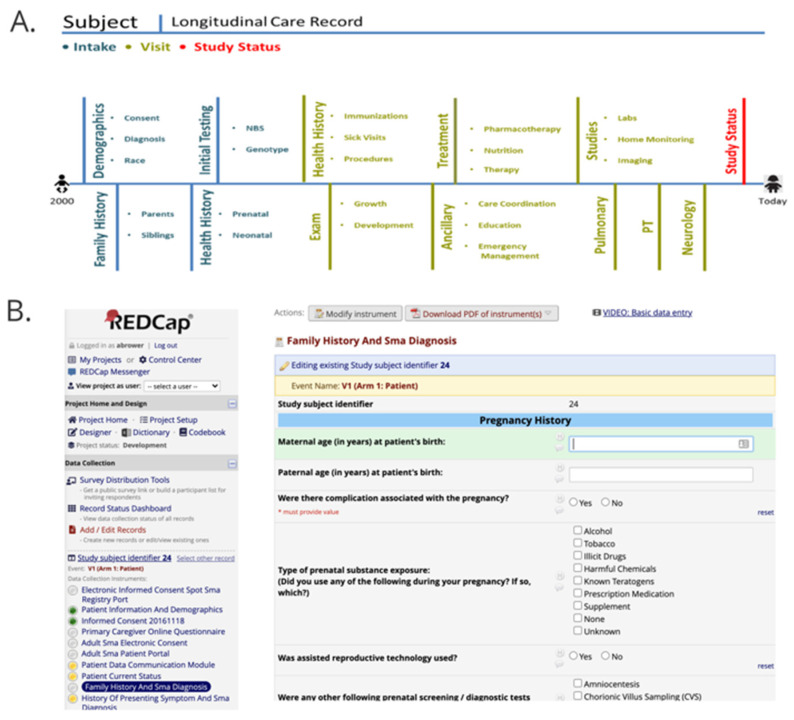
(**A**) Subject’s Longitudinal Care Timeline and (**B**) REDCap Electronic Case Report Form.

**Figure 3 IJNS-07-00037-f003:**
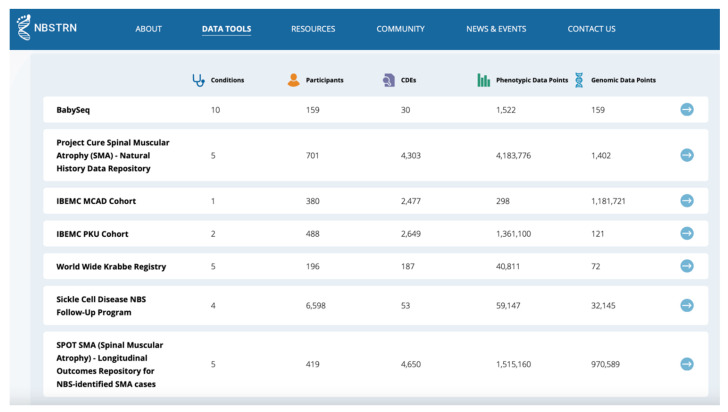
LPDR Projects Available for Secondary Use.

**Figure 4 IJNS-07-00037-f004:**
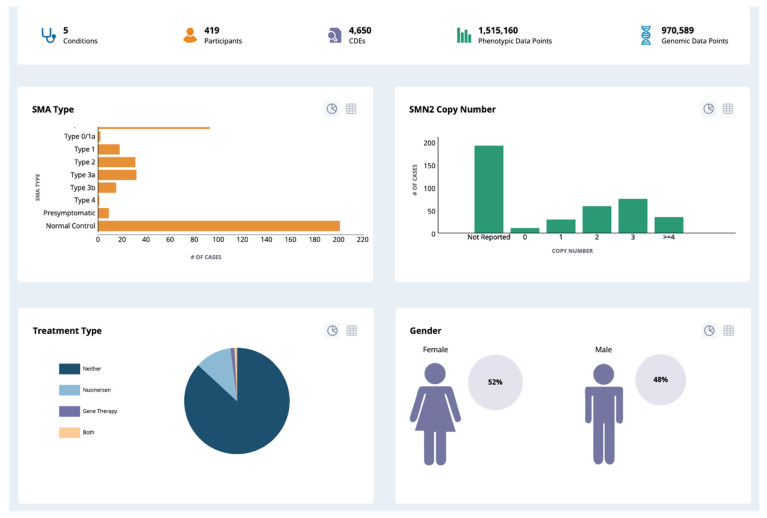
LPDR Data Visualization.

**Table 1 IJNS-07-00037-t001:** Summary of Projects in the NBSTRN LPDR.

Project	Condition(s)	Population	CDEs	Subjects	Max Data Entry Time Points	User Type	Data Available? * (Yes/No)	Status
Sickle Cell Disease Newborn Screening Follow-Up	Sickle Cell Disease	Diagnosed Cases	57	8346	9	Community-Based Organization (CBO)	Yes	Active
Inborn Errors in Metabolism Consortium (IBEMC)	43 RUSP Conditions **	Diagnosed Cases	2649	2078	30	Clinician	Yes	Completed
Project CURE Spinal Muscular Atrophy (SMA)	SMA	Diagnosed Cases	4337	701	5	Clinician	Yes	Active
SPOT SMA	SMA	Newborns	4759	434	4	Clinician	Yes	Active
Krabbe Registry	Krabbe	Diagnosed Newborns	187	196	1	Clinician and Parent	Yes	Completed
Rady NSIGHT2	Variety	Neonatal Intensive Care Unit (NICU)	45	214	2	Clinician	Yes	Completed
BabySeq	Variety	NICU; Healthy at Birth	30	159	2	Clinician	Yes	Completed
NC NEXUS	RUSP ***	Newborns	180	200	2	Clinician	Yes	Completed
NY State LTFU	X-linked Adrenoleukodystrophy, Severe Combined Immune Deficiency, Fatty Acid Oxidation Disorders, Krabbe	Newborns	202	10	5	State NBS Program	No	Active
Duchenne NBS Pilot	Duchenne Muscular Dystrophy	Newborns	624	17	6	Clinician and State NBS Program	No	Active
eXtraordinarY Babies Study	Chromosome Aneuploidy	Newborns	320	200	5	Clinician	No	Active
CPT1A Arctic Variant	CPT1A Arctic Variant	Newborn and Children	69	TBD	4	Clinician	No	Active
Oregon Health & Science University	RUSP ***	Newborns	894	TBD	2	Clinician	No	Active

* Data availability subject to approval from the data submitter/owner. ** 2-Methyl-3-hydroxybutyric aciduria, 2-Methylbutyrylglycinuria, 3-Methylcrotonyl-CoA carboxylase deficiency, 3-Methylglutaconic aciduria, Argininosuccinic aciduria, Disorders of biopterin biosynthesis, Disorders of biopterin regeneration, Biotinidase deficiency, beta-Ketothiolase deficiency, Carnitine-acylcarnitine translocase deficiency, Methylmalonic acidemia (Cobalamin disorders), Methylmalonic acidemia (Cobalamin disorders), Primary Congenital Hypothyroidism, Citrullinemia type I, Citrullinemia type II, Carnitine palmitoyl transferase I deficiency, Carnitine palmitoyl transferase II deficiency, Carnitine uptake defect, Glutaric acidemia type I, Glutaric acidemia type II, Galactoepimerase deficiency (uridine diphosphate galactose 4-epimerase deficiency), Galactokinase deficiency, Classical galactosemia (galactose-1-phosphate uridyltransferase deficiency), Hyperphenylalaninemia (variant, benign), Homocystinuria, 3-Hydroxy-3-methylglutaric aciduria, Isobutyrylglycinuria, Isovaleric acidemia, Malonic acidemia, Medium-chain acyl-CoA dehydrogenase deficiency, Multiple carboxylase deficiency, Hypermethioninemia, Maple syrup urine disease, Methylmalonic acidemia (methylmalonyl-CoA mutase), Phenylketonuria, Proionic acidemia, Short-chain acyl-CoA dehydrogenase deficiency, Trifunctional protein deficiency, Tyrosinemia type I, Tyrosinemia type II, Tyrosinemia type III, Very long-chain acyl-CoA dehydrogenase deficiency. *** https://www.hrsa.gov/advisory-committees/heritable-disorders/rusp/index.html (accessed on 12 April 2021).

## Data Availability

The data presented in this study are openly available in https://nbstrn.org/tools/lpdr (accessed on 29 June 2021).

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
