# Peer review of "The Longitudinal Pediatric Data Resource: Facilitating Longitudinal Collection of Health Information to Inform Clinical Care and Guide Newborn Screening Efforts"

_2409-515X, 2021, doi:10.3390/ijns7030037_

Round 1
Reviewer 1 Report
This article describes the development of a data resource to facilitate longitudinal collection of health information for follow up of individuals identified by newborn screening with a confirmed condition. It is unique in that the resources were not previously available to begin to address the long term outcomes of interventions that newborn screening affords affected individuals over the lifespan. The approach included the identification of common data elements important to track for multiple rare diseases and appropriate privacy controls for access to the data base for researchers and clinicians. The article also includes current and past projects that have utilized the data base illustrating the value of this approach. This information is critical in improving the newborn screening system to improve the care of those individuals identified by newborn screening.
Author Response
Thank you for the helpful review and comments. We agree with your points regarding the uniqueness of the resource, the attention to privacy controls, and the usefulness of past and current projects that utilize the resource. We performed a spell check as requested and updated the manuscript to reflect these changes.
Reviewer 2 Report
This manuscript describes the structure and potential values of the LDPR, a complex and likely underused resource that is important to the readership of this journal.
Only minor suggestions:
- line 69: The sentence that begins with "While..." is a long run on sentence that is incomplete. At a minimum, the word "while" could be removed and the sentence could start with "The...", but the authors might also consider breaking this sentence up into a couple of shorter sentences.
- I could not find the text associate with FIgure 3 - FIgure 3 not mentioned. Perhaps this should be referenced at the end of the sentence in line 231
- Line 274 - Figure # should likely be Figure 4.
- Figure 1 takes up a lot of space with the pictures - these are not necessary, and the whole figure is expendable. The authors could use the full component names in the figure in the referring sentence in line 42. Alternatively, a version of the elements in Figure 1 could be combined with Figure 2.
- More useful than Figure 1, would be a figure that demonstrated a set of CDEs for a disorder and what a REDCap data entry form might look like - this could be referred to from section 3.1
Author Response
- line 69: The sentence that begins with "While..." is a long run on sentence that is incomplete. At a minimum, the word "while" could be removed and the sentence could start with "The...", but the authors might also consider breaking this sentence up into a couple of shorter sentences.
- Reviewer's suggestion noted and sentence broken up into two and revised as follows: "The ability to capture clinical information early in the clinical course of a disease, even before clinical symptoms appear, advances disease understanding. Longitudinal data collection also helps to establish the efficacy of new treatments and management approaches, informs the community at large about the value of early identification and treatment through newborn screening, and identifies areas for improvement in disease management throughout the lifespan."
- I could not find the text associate with FIgure 3 - FIgure 3 not mentioned. Perhaps this should be referenced at the end of the sentence in line 231
- The reviewer is correct that a reference to Figure 3 is missing. It was added as follows on line 231 - Note that Figure 3 is now Figure 2 based on removal of Figure 1 at reviewer's suggestion "The LPDR has data displays that describe the types of data and populations available for secondary use (Figure 2)."
- Line 274 - Figure # should likely be Figure 4.
- The reviewer is correct that line 274 refers to Figure 4 and the text was modified as follows and note that Figure 4 is now Figure 3 based on removal of Figure 1 at reviewer's suggestion - "To facilitate the utilization and application of LPDR, NBSTRN has displayed a set of LPDR studies and data visualization on the website www.nbstrn.org (see Figure 4)."
- Figure 1 takes up a lot of space with the pictures - these are not necessary, and the whole figure is expendable. The authors could use the full component names in the figure in the referring sentence in line 42. Alternatively, a version of the elements in Figure 1 could be combined with Figure 2.
- We agree with the reviewer's suggestion and removed Figure 1. We updated Figure 2 to more concisely communicate the 3 key features and 4 components included in the ACHDNC LTFU statement.
- More useful than Figure 1, would be a figure that demonstrated a set of CDEs for a disorder and what a REDCap data entry form might look like - this could be referred to from section 3.1
- We agree with the reviewer's suggestion and have drafted and inserted a new figure as described.